# Molecular-strain induced phosphinidene reactivity of a phosphanorcaradiene

Yizhen Chen[1,2,4], Peifeng Su[3,4], Dongmin Wang[1,2], Zhuofeng Ke[3] ✉ & Gengwen Tan[1,2] ✉

Phosphanorcaradienes are an appealing class of phosphorus compounds that can serve as synthons of transient phosphinidenes. However, the synthesis of such species is a formidable task owing to their intrinsic high reactivity. Herein we report straightforward synthesis, characterization and reactivity studies of a phosphanorcaradiene, in which one of the benzene rings in the flanking fluorenyl substituents is intramolecularly dearomatized through attachment to the phosphorus atom. It is facilely obtained by the reduction of phosphorus(III) dichloride precursor with potassium graphite. Despite being thermally robust, it acts as a synthetic equivalent of a transient phosphinidene. It reacts with trimethylphosphine and isonitrile to yield phosphanylidene-phosphorane and 1-phospha-3-azaallene, respectively. When it is treated with one and two molar equivalents of azide, iminophosphane and bis(imino)phosphane are isolated, respectively. Moreover, it is capable of activating ethylene and alkyne to afford [1 + 2] cycloaddition products, as well as oxidative cleavage of Si−H and N−H bonds to yield secondary phosphines. All the reactions proceed smoothly at room temperature without the presence of transition metals. The driving force for these reactions is most likely the high ring-constraint of the three-membered $PC_2$ ring and recovery of the aromaticity of the benzene ring.

Carbene insertion into aromatic rings to produce cycloheptatrienes (CHTs) and norcaradienes (NCDs) developed by Büchner and Curtius in 1885 is a powerful method for the conversion of stable aromatic compounds to more reactive systems (Fig. 1a)[1]. The equilibria between CHTs and NCDs have been well-established[2]. Recent reports have shown that CHTs and NCDs can serve as carbene precursors via retro-Büchner reactions[3,4]. Phosphorus is viewed as carbon copy[5]. Phosphinidenes are neutral monocoordinate and monovalent phosphorus species possessing six valence electrons. Such species are isoelectronic to well-explored carbenes[6–14] and tetrylenes[15–17]. In contrast, most phosphinidenes are transient species because of their exceptionally high reactivity[18–21]. The sole isolable example of phosphinidene

was synthesized by Bertrand and coworkers[22]. The chemical properties of phosphinidenes have been mainly demonstrated by generating transient species with suitable precursors.

Phosphepines, phosphorus analogs of CHTs, were first synthesized by Märkl et al. in the 1980s[23,24], followed by the groups of Tsuchiya[25,26] and Lammertsma[27]. The Lammertsma group showed that phosphepine transition metal (TM) complex **I** could act as a versatile metallo-phosphinidene precursor to activate a variety of substrates (Fig. 1b)[28–31]. In addition, several P(III) precursors, including phosphatetrahdrane **II**[32], 7-phosphanorbornadiene **III**[33], dibenzo derivatives **IV**[34–36] and phosphirane-TM complexes[37–41], have been utilized for phosphinidene reactivity studies. Notably, phosphinidene transfer

[1]Key Laboratory of Bioinorganic and Synthetic Chemistry of Ministry of Education, Guangdong Basic Research Center of Excellence for Functional Molecular Engineering, School of Chemistry, IGCME, Sun Yat-sen University, Guangzhou 510275, China. [2]Innovation Center for Chemical Sciences, Key Laboratory of Organic Synthesis of Jiangsu Province, College of Chemistry, Chemical Engineering and Materials Science, Soochow University, Suzhou 215123, China. [3]School of Materials Science and Engineering, PCFM Lab, the Key Laboratory of Low-carbon Chemistry & Energy Conservation of Guangdong Province, Sun Yat-sen University, Guangzhou 510006, China. [4]These authors contributed equally: Yizhen Chen, Peifeng Su. ✉e-mail: kezhf3@mail.sysu.edu.cn; tangw55@mail.sysu.edu.cn

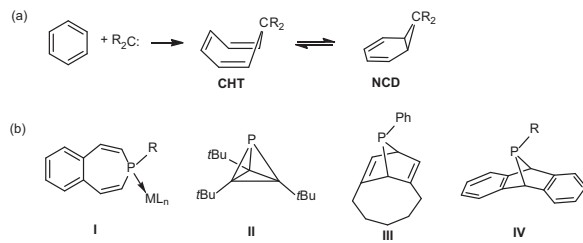

**Fig. 1 | Büchner ring expansion and selected P(III) compounds that act as synthons of transient phosphinidenes. a** Schematic depiction of Büchner ring expansion and the equilibria between cycloheptatriene (CHT) and norcaradiene (NCD); R is organic substituent. **b** Selected examples of P(III) precursors as synthons of transient phosphinidenes; R is organic substituent, and L is coordinating ligand.

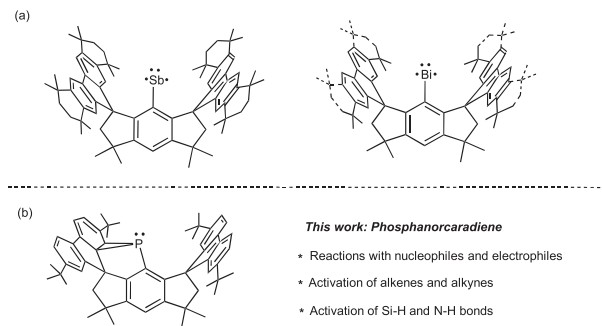

**Fig. 2 | Triplet pnictinidenes and phosphanorcaradiene supported by hydrindacene ligands. a** Triplet stibinidene and bismuthinidenes; (**b**) phosphanorcaradiene in this work.

with these materials must be promoted through heating or via TM catalysts.

In contrast to phosphepines, phosphanorcaradienes, which are phosphorus analogs of NCDs, have higher molecular-constraints and thereby exhibit inherent high reactivity, making their isolation in the condensed state a formidable task. On this basis, such compounds may be ideal synthons for accessing transient phosphinidenes via phosphorus-type retro-Büchner reactions. However, only one free phosphanorcaradiene has been reported by Stephan and coworkers. It was synthesized through demetalation of a phosphepine-gold complex, while its reactivity as a phosphinidene precursor for small molecule activation has not been fully disclosed[42]. Very recently, a ruthenophosphanorcaradiene acting as a synthon for an ambiphilic metallophosphinidene was reported by Scheer, Tilley and coworkers[43].

Our group has continuous research interests in synthesizing low-coordinate main-group species, and has successfully isolated and structurally characterized several heavier analogs of free carbynes[44–47], triplet stibinidene and bismuthinidenes (Fig. 2a)[48–50] supported by sterically encumbered hydrindacene ligands[51]. Encouraged by these results, we continued our research to pursue isolable phosphinidenes. In this contribution, we report straightforward synthesis, characterization and reactivity studies of a phosphanorcaradiene **1** (Fig. 2b). Reactivity studies reveal that it can serve as an elegant synthon of phosphinidene due to the release of molecular-strain[52,53].

## Results

### Synthesis and characterization of 1

With the aim of synthesizing a stable phosphinidene, we carried out the reduction of the phosphorus(III) dichloride M$^s$Fluid$^{tBu}$-PCl$_2$[54] with two molar equivalents of potassium graphite in THF; however, phosphanorcaradiene **1** was obtained in 75% yield as a yellow solid instead

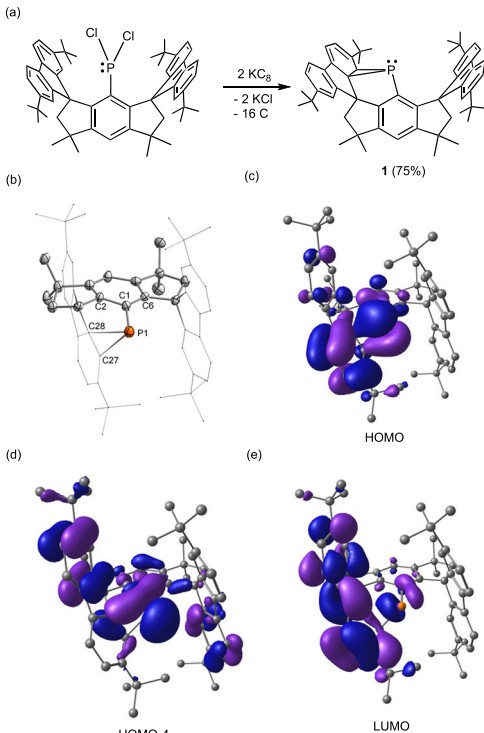

**Fig. 3 | Synthesis, molecular structure and selected frontier molecular orbitals of phosphanorcaradiene 1. a** Reduction of phosphorus(III) dichloride precursor with two molar equivalents of potassium graphite. **b** Thermal ellipsoid drawings of the molecular structure of **1** at 50% probability. All hydrogen atoms are omitted, and the fluorenyl moieties are shown in a wireframe style for clarity. Selected bond lengths (Å) and angles (°): P1–C1 1.847(4), P1–C27 1.973(5), P1–C28 1.985(5), C27–C28 1.476(6); C1–P1–C27 96.4(2), C1–P1–C28 88.06(19), C27–P1–C28 43.78(19), C27–C28–P1 67.7(3). **c** HOMO. **d** HOMO-4; (**e**) LUMO.

of the expected product (Fig. 3a). One of the benzene rings in the flanking fluorenyl moieties is intramolecularly dearomatized through attachment to the phosphorus atom. Contrastingly, similar reduction reactions with less sterically hindered ligands afforded diphosphenes or other higher oligomers[55–57]. The formation of the PC$_2$ three-membered ring leads to a decrease in the symmetry and complex $^1$H and $^{13}$C{$^1$H} NMR spectra. The proton signal in the PC$_2$ ring was shown at δ 2.54 ppm. A sharp singlet signal at δ −155.1 ppm was observed in the $^{31}$P{$^1$H} NMR spectrum in C$_6$D$_6$ solution, similar to that of phosphiranes[22,58]. Interestingly, **1** can be heated to 100 °C in C$_6$D$_6$ solution for one hour without noticeable decomposition under an inert atmosphere, but it is highly air- and moisture-sensitive and yields intractable mixtures when exposed to air or dry oxygen. The synthesis of **1** is more straightforward in comparison to that of the phosphanorcaradiene reported by Stephan and coworkers.[18]

Single crystals of **1** suitable for X-ray diffraction analysis were obtained by layering n-hexane in a toluene solution at 4 °C[59]. It crystallizes in the triclinic space group $P\bar{1}$. There are two independent enantiomers present in one crystal unit cell due to the chirality of the P atom. One of the molecules is shown in Fig. 3b, which unambiguously reveals the connection of the P atom to two C atoms of the flanking fluorenyl group. The distances of P1–C27 (1.973(5) Å) and P1–C28 (1.985(5) Å) are substantially longer than those of P1–C1 (1.847(4) Å) and the P–C distances of the PC$_2$ ring in phosphiranes[22,58]. Consistently, the C27-P1-C28 angle (43.78(19)°) is more acute. Moreover, the P atom deviates from the position expected for atoms attached to a phenyl group as evidenced by the large difference between the bond angles of C2-C1-P1 (111.7(3)°) and C6-C1-P1 (131.7(4)°). These data suggest that the PC$_2$ ring in **1** bears a greater ring-strain than phosphiranes.

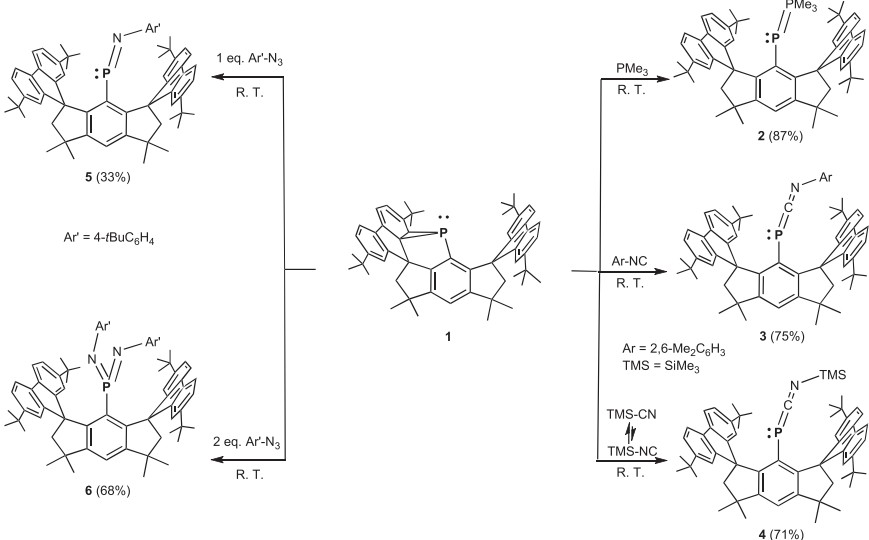

**Fig. 4 | Reactions of 1 with nucleophiles and electrophiles.** Treatment of **1** with trimethylphosphine afforded phosphanylidene-phosphorane **2**. Reactions of **1** with 2,6-dimethylphenylisonitrile and trimethylsilylnitrile yielded 1-phospha-3-azaallenes **3** and **4**, respectively. **1** reacted with one and two molar equivalents of 4-tertbutylphenylazide afforded iminophosphorane **5** and bis(imino)phosphorane **6**, respectively.

## Theoretical calculations of 1

The electronic structure of **1** was further elucidated by density functional theory (DFT) calculations at the BP86 + D3BJ/def2-SVP level. The HOMO represents the bonding orbitals of the $C_2P$ ring and the C–C π-orbitals of the linked $C_6$ ring (Fig. 3c–e). The lone pair of P atom is shown in the HOMO-4, and the LUMO contains the π*-antibonding orbitals of the fluorenyl moiety linked to the P atom. Additionally, natural bond orbital (NBO) and intrinsic bond orbital (IBO)[60] analyses were conducted to gain insight into the bonding character of **1**. The Wiberg bond indices (WBIs) of the P1–C27 (0.74) and P1–C28 (0.72) bonds are substantially smaller than that of the P1–C1 bond (0.96), suggesting that they are relatively weaker than the P1–C1 bond. Moreover, they are smaller than those of the phosphanorcaradiene reported by Stephan and coworkers (0.79 and 0.76), implying a more fragile $PC_2$ ring in **1**. IBO analysis shows that P1–C27(C28) σ-bonds are formed by the overlap of the 3p orbitals at P1 and the 2p orbitals at C27(C28) (Supplementary Fig. 1). The 3p orbitals at the P atom and the 2p orbitals at the C atoms provided more than 90% of the bonding electrons in the P1–C27(C28) bonds. These orbitals are significantly different from the P1–C1 σ-bond, which is formed by the 3p orbital with the partial s character of P1 and the $sp^2$ hybrid orbital of C1.

To further explore the electronic structure of the $C_2P$ ring of **1**, the quantum theory of atoms in molecules (QTAIM) and electron localization function (ELF) analyses were carried out[61,62]. The QTAIM calculations indicate that there are three bond critical points (BCPs) among the P1, C27 and C28 atoms (Supplementary Fig. 2). In addition, the bond paths between P1 and C27(C28) slightly deviate away from the P1–C27(C28) axes. Furthermore, the ELF plots exhibit a twisted region with highly localized electron density between each C and P in the $C_2P$ ring (Supplementary Fig. 3).

The formation of **1** can be simplified to [2 + 1] cycloaddition of a transient phosphinidene with a C = C double of the flanking fluorenyl group. We previously reported the isolation of triplet stibinidene[48] and bismuthinidenes[49], which are heavy congeners of the proposed phosphinidene. A comparison of the calculated energies between triplet phosphinidene and **1** showed that **1** is 5.7 kcal/mol lower in energy than that of triplet phosphinidene (Supplementary Fig. 4). The conversion of triplet phosphinidene to **1** is facile with a predicted barrier no more than 5 kcal/mol.

To provide further understanding of **1**, electronic and steric properties were theoretically studied. The proton affinity was analyzed to evaluate the electronic property, and meanwhile related calculations on PPh₃ and Mes-P(CH₂CH₂) were also performed for comparison. As shown in Supplementary Fig. 5, the proton affinity of **1** is −156.8 kcal/mol, which is lower than those of PPh₃ (−152.8 kcal/mol) and Mes-P(CH₂CH₂) (−141.8 kcal/mol). Compared with PPh₃ and the analogous three-membered ring phosphirane Mes-P(CH₂CH₂), **1** has a better capability of electron donating. In addition, the steric property of **1** were studied with percent buried volume (%$V_{bur}$) analysis. The %$V_{bur}$ of **1** is 76.6%, higher than those of PPh₃ (65.0%) and Mes-P(CH₂CH₂) (54.3%). The results of percent buried volumes suggest that the steric hindrance around the phosphorus atom of **1** is greater than those of PPh₃ and Mes-P(CH₂CH₂). Besides, steric maps shown in Supplementary Fig. 5 also display a larger steric hindrance of **1** than PPh₃ and Mes-P(CH₂CH₂). Owing to the steric effect, this active phosphanorcaradiene species **1** can be stabilized.

## Reactions of 1 with nucleophiles and electrophiles

With understanding of the electronic structure, it is anticipated that **1** may act as a phosphinidene synthon through the dissociation of relatively weak P1–C27(C28) bonds and the release of molecular-strain. Accordingly, the reactivity of **1** toward nucleophiles and electrophile was tested (Fig. 4). It reacted smoothly at room temperature with trimethylphosphine to afford phosphanylidene-phosphorane **2**, which was isolated in 87% yield. **2** exhibits ³¹P NMR resonance signals at δ −0.9 and −157.8 ppm with a coupling constant of $^1J_{P\text{-}P}$ = 603 Hz. The high-field signal is comparable to those of reported phosphanylidene-phosphoranes[63,64], which have been utilized as phospha-Wittig reagents[65–67]. Moreover, treatment of **1** with 2,6-dimethylphenylisonitrile and trimethylsilylnitrile both afforded 1-phospha-3-azaallenes **3** and **4** in high yields, respectively. ³¹P resonances were observed at δ −152.3 and −190.5 ppm in the ³¹P{¹H} NMR spectra. The formation of **4** can be interpreted by the equilibrium between trimethylsilylnitrile and trimethylsilylisonitrile[68], and the latter species has a greater propensity to react with **1**. A similar reaction pattern between disilyne and trimethylsilylnitrile has been reported by Sekiguchi and coworkers[69]. Additionally, **1** reacted with one and two molar equivalents of 4-tertbutylphenylazide yielded iminophosphorane **5** and bis(imino)

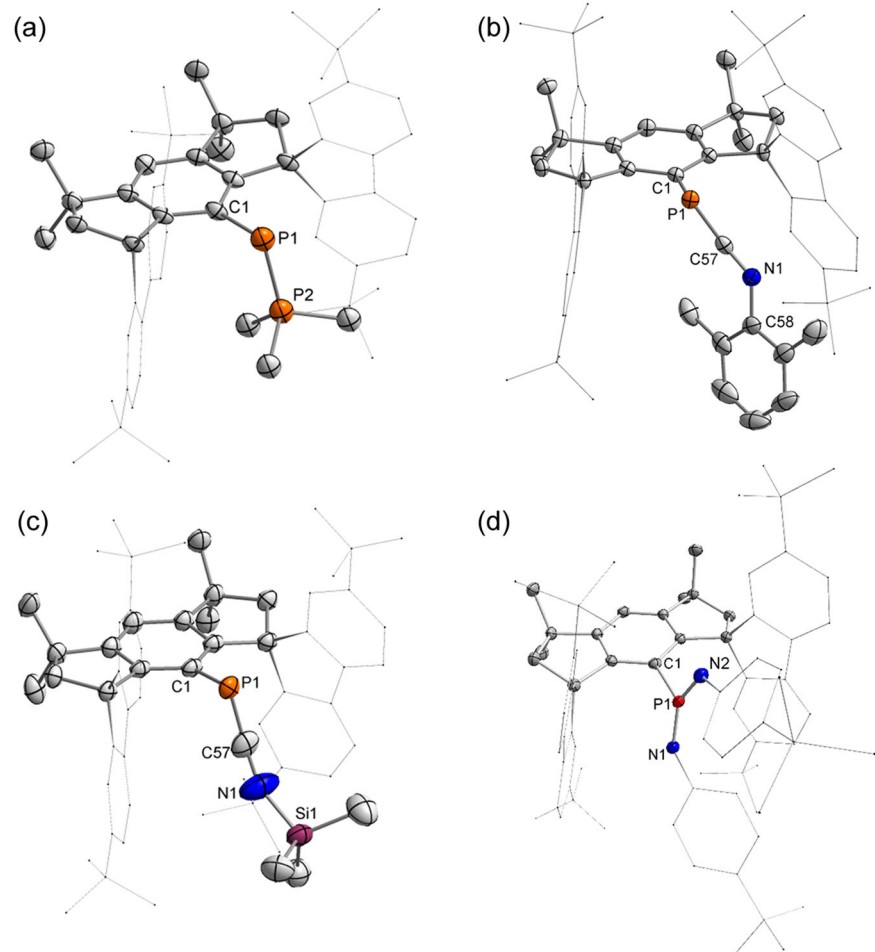

**Fig. 5 | Molecular structures of 2-4 and 6.** Thermal ellipsoid drawings of the molecular structures of **2** (**a**), **3** (**b**), **4** (**c**) and **6** (**d**) at 50% probability. All hydrogen atoms are omitted, and the fluorenyl moieties and the 4-tBuC₆H₄ groups are shown in a wireframe style for clarity.

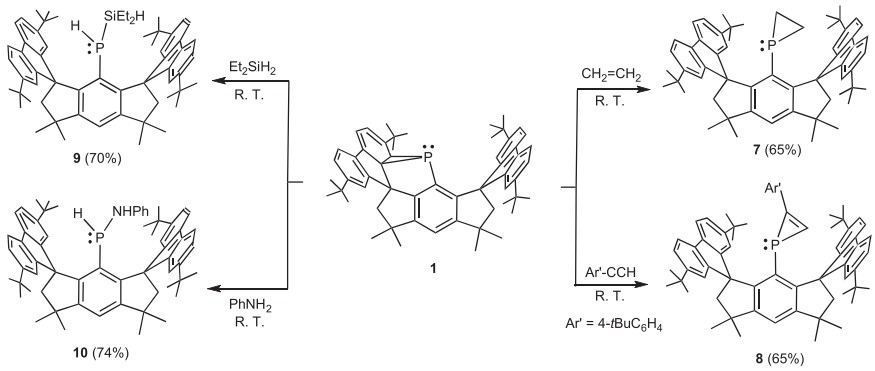

**Fig. 6 | Activations of unactivated alkene, alkyne, silane and amine with 1.** The reaction of 1 with ethylene gave phosphirane **7**; **1** reacted with 4-tertbutylphenylacetylene to afford phosphirene **8**; **1** activated Si–H and N–H bonds to yield phosphines **9** and **10**, respectively.

phosphorane **6**, respectively. Their $^{31}$P{$^{1}$H} NMR spectra show singlet signals at δ 424.2 and 45.1 ppm, respectively.

The dicoordinate phosphorus atoms in **2-4** were revealed by SC-XRD analysis (Fig. 5). The P–P distance of 2.1306(15) Å and the C1-P1-P2 bond angle of 102.03(11)° in **2** are comparable to those in DmpP=PMe₃ (Dmp = 2,6-Mes₂C₆H₃; 2.084(2) Å and 106.79(13)°, respectively)[70]. The P = C = N fragments in **3** and **4** are almost linear with P-C-N bond angles of 168.82(18)° and 168.0(3)°, respectively. Moreover, the P1-C57 (1.668(2) and 1.651(4) Å) and C57-N1 (1.200(3) and 1.212(5) Å) bonds feature double bond characteristics. These geometric parameters are

reminiscent of (phosphino)phosphinidene-isonitriles reported by Bertrand and coworkers[71], and are consistent with their cumulene-like properties. SC-XRD analysis shows that the phosphorus atom in **6** exhibits a trigonal planar geometry with a sum of angles around the P1 atom of 360°. Moreover, the distances of P1-N1 (1.5408(11) Å) and P1-N2 (1.5420(11) Å) suggest their double bond characters.

**Reactions of 1 with alkene and alkyne**

The ability of **1** to activate alkene and alkyne was further investigated (Fig. 6). Exposure of **1** to 1 atm. of ethylene atmosphere at room

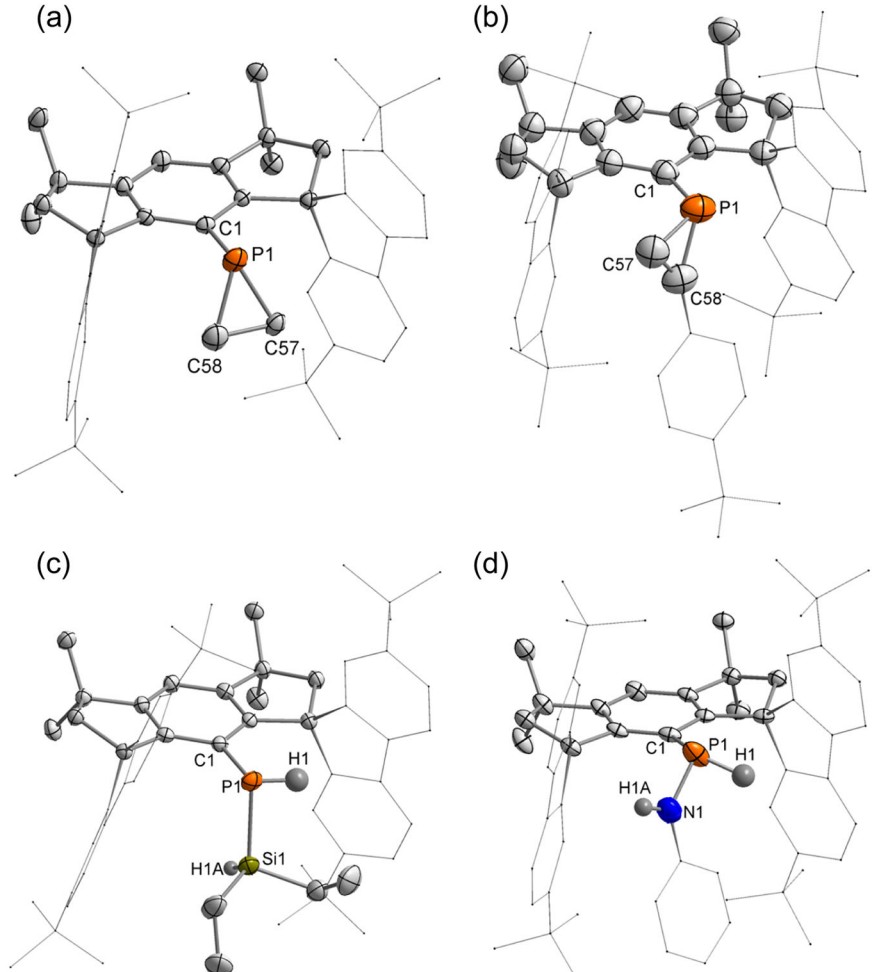

**Fig. 7 | Molecular structures of 7-10.** Thermal ellipsoid drawings of the molecular structures of **7** (**a**), **8** (**b**), **9** (**c**) and **10** (**d**) at 50% probability. All hydrogen atoms except those at the P, Si and N atoms in **7** and **8** are omitted, and the fluorenyl moieties, the 4-tBuC₆H₄ and phenyl groups are shown in a wireframe style for clarity.

temperature led to the formation of phosphirane **7** in moderate yield. The $^{31}$P{$^{1}$H} NMR resonance was shown at δ −229.2 ppm, similar to that of 1-mesitylphosphirane (δ −238.9 ppm)[72]. Complete consumption of **1** with an excess amount of 4-tertbutylphenylacetylene at room temperature was observed after 12 h, and phosphirene **8** was isolated in 65% yield. A single resonance signal was observed at δ −171.3 ppm in the $^{31}$P{$^{1}$H} NMR spectrum of **8**. The formation of **7** and **8** under mild conditions is most likely attributed to the decrease in the ring-strain and recovery of aromaticity at the six-membered ring in **1**, which is striking since phosphinidene transfer to alkenes with **II** and **IV** developed by the Cummins group has to be promoted by heating or catalyzed by TM catalysts[32,34,35]. It is noteworthy that **7** and **8** do not react with 4-tertbutylphenylazide, most probably attributed to high steric hindrance around the phosphorus atoms.

The molecular structures of **7** and **8** determined by SC-XRD analysis are shown in Fig. 7. New three-membered PC₂ rings are shown in the structures. The C57−C58 distances are 1.513(4) and 1.308(5) Å, respectively, in accordance with their single and double bond nature. Moreover, the P−C bond lengths (1.847(3) and 1.848(3) Å for **7**; 1.779(4) and 1.808(4) Å for **8**) inside the PC₂ ring are shortened in comparison to those in **1**. In contrast to that of **1**, the P atoms in **7** and **8** are in the expected positions for atoms attached to a phenyl ring.

**Activation of Si−H and N−H bonds with 1**

Recently, geometrically constrained phosphines have shown to be capable of σ-bond activation[73–76], prompting us to study the ability of **1** in σ-bond activation. Interestingly, we found that **1** could activate the Si−H and N−H bonds. The reactions of **1** with diethylsilane and phenylamine occurred smoothly at room temperature, and silylphosphine **9** and aminophosphine **10** were obtained in 70% and 74% yields, respectively (Fig. 6). Their structures were unambiguously determined by SC-XRD analysis (Fig. 7). The activation of Si−H and N−H bonds represents rare examples of intermolecular inert bond activation with metal-free transient phosphinidenes. Oxidation cleavage of Si−H bonds with electrophilic metallo-phosphinidene complexes was reported by Sterenberg et al.[77,78]. Stephan and coworkers showed hydrosilylation of low-valent P₅Ph₅ mediated by the Lewis acid B(C₆F₅)₃[79]. Hering-Junghans and coworkers recently revealed the first example of N−H activation at a metal-free P(I) center with phospha-Wittig reagents[66]. Additionally, the reactions of **1** with dioxygen and dihydrogen led to intractable mixtures.

**Theoretical studies of the reaction mechanisms**

The mechanisms for the activation of alkene, alkyne, silane, and amine with **1** were studied by DFT calculations. For alkene, alkyne, and silane, as shown in Fig. 8a–c, respectively, the breakage of the PC₂ three-membered ring structures and activations of the C = C bond, the C ≡ C bond, and the Si−H bond occur concertedly in respective transition states (**7-TS**, **8-TS**, and **9-TS**, respectively). Through respective transition states, the addition products **7**, **8**, and **9** are formed. While for the activation of amine, the phosphorus atom interacts with amine to form **10-RC** firstly, in which the three-membered ring structure in **1** is broken

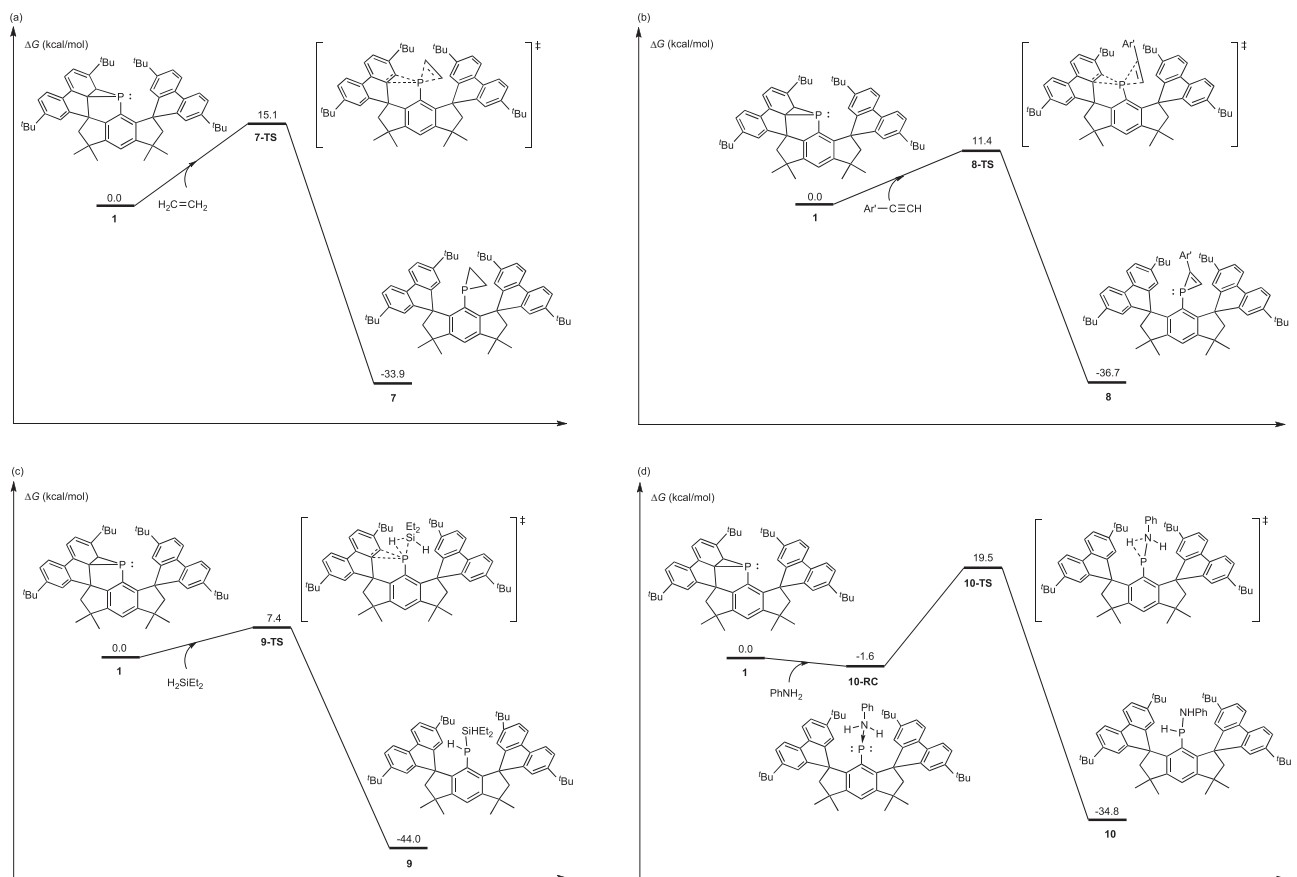

**Fig. 8 | Studies of the reaction mechanisms through DFT calculations.** Gibbs free energy profiles of activations of (**a**) alkene, (**b**) alkyne, (**c**) silane, and (**d**) amine with **1** (Ar' = 4-$^t$BuC$_6$H$_4$, Gibbs free energies are shown in kcal/mol). Calculations were performed at the BP86-D3/def2-TZVPP//BP86-D3/def2-SVP level of theory.

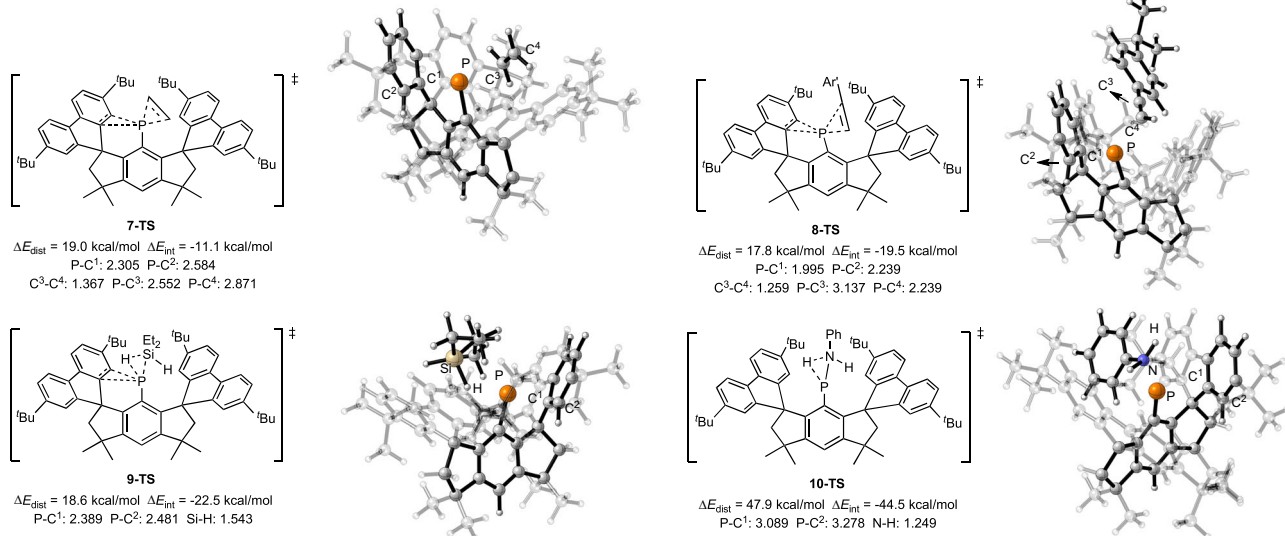

**Fig. 9 | Theoretical investigation of the structures of the transition states.** Structures of transition states in activations of alkene, alkyne, silane, and amine (distance are shown in Å). Calculations were performed at the BP86-D3/def2-TZVPP//BP86-D3/def2-SVP level of theory.

(Fig. 8d). The N–H bond is activated through **10-TS** to form product **10**. The energy barriers for activations of alkene, alkyne, silane, and amine are 15.1, 11.4, 7.4, and 21.1 kcal/mol, respectively, which are in line with the reaction conditions of room temperature. As shown in Fig. 8, these transformations are exothermic processes through early transition states, and thus the activations of alkene, alkyne, silane, and amine are facile by the active phophanorcaradiene **1**.

In addition, the structures of **7-TS**, **8-TS**, **9-TS**, and **10-TS** were analysed. As Fig. 9 shows, P–C$^1$ and P–C$^2$ distances in these transition states are all larger than those in **1** (1.980 and 2.002 Å for P-C$^1$ and P-C$^2$ respectively), indicating the breakage of the three-membered ring structure. The C$^3$–C$^4$ distance in **7-TS** is 1.367 Å, and the C$^3$–C$^4$ distance in **8-TS** is 1.259 Å. These C$^3$–C$^4$ distances respectively suggest that the C = C bond and the C ≡ C bond are activated. Besides, **7-TS** and **8-TS** are

typical concerted asynchronous transition states. For **7-TS**, the P–C$^3$ and the P–C$^4$ distances are 2.552 and 2.871 Å, respectively, and for **8-TS**, the P–C$^3$ and the P–C$^4$ distances are 3.137 and 2.239 Å respectively. For **9-TS**, the Si–H distance is 1.543 Å, and the N–H distance in **10-TS** is 1.249 Å, suggesting the activations of the Si–H bond and the N–H bond. Furthermore, distortion analysis was performed on **7-TS, 8-TS, 9-TS**, and **10-TS**[80,81]. In the distortion analysis, these transition states were divided into the activated molecule moiety and the phophanorcaradiene moiety. The total deformation energies ($\Delta E_{dist}$) of **7-TS, 8-TS, 9-TS**, and **10-TS** are shown in Fig. 9. The deformations of **7-TS, 8-TS**, and **9-TS** compared to respective **RC**s are not as large as that of **10-TS**, probably due to the aromatization driving force of the decyclization of P-ligand three-membered ring structure in the transition state. The $\Delta E_{dist}$ of **7-TS, 8-TS**, and **9-TS** are 19.0, 17.8, and 18.6 kcal/mol respectively, which are all lower than that of **10-TS** (47.9 kcal/mol). Therefore, the energy barrier of the activation of amine is relatively higher than those of activations of alkene, alkyne, and silane. In addition, the interaction energies ($\Delta E_{int}$) of transition states were also analyzed. $\Delta E_{int}$ of **10-TS** is −44.5 kcal/mol, which is much higher than those of **7-TS, 8-TS**, and **9-TS** (−11.1, −19.5, and −22.5 kcal/mol, respectively). Although the $\Delta E_{dist}$ of **10-TS** is high, the high $\Delta E_{int}$ of **10-TS** leads to a relatively low energy barrier for amine activation.

## Discussion

In summary, we have described the synthesis and characterization of thermally robust phosphanorcaradiene **1** with high molecular-strain. It could react with nucleophiles and electrophiles. Strikingly, **1** is capable of activating C = C double and C ≡ C triple bonds, as well as Si–H and N–H bonds at room temperature. The driving force for these reactions is most likely the high ring-strain of the three-membered PC$_2$ ring and recovery of aromaticity of the C$_6$ ring. This work demonstrates the great potential of phosphanorcaradiene as a synthetic equivalent of phosphinidene in small molecule activation. The use of **1** to synthesize more phosphorus compounds including phosphinidene-TM catalysts is under investigation in our laboratory.

## Methods

All experiments were carried out under a dry oxygen-free nitrogen atmosphere using standard Schlenk techniques or in a N$_2$ filled-glove box. Solvents were dried by standard methods and stored in activated 4 Å molecule sieve in the glovebox. All reagents were purchased from commercial sources (Energy Chemical and TCI) and used without further purification unless otherwise noted. M$^s$Fluind$^t$Bu-PCl$_2$[54], and KC$_8$[82] were synthesized according to reported procedures. The $^1$H, $^{13}$C{$^1$H} and $^{31}$P{$^1$H} NMR spectra were recorded on Bruker spectrometers (AV400 and AV600). Chemical shift values for protons are referenced to the residual proton resonance of CDCl$_3$ (δ: 7.26), C$_6$D$_6$ (δ: 7.16), THF-$d_8$ (δ: 3.62); chemical shift values for carbons are referenced to the carbon resonance of CDCl$_3$ (δ: 77.16), C$_6$D$_6$ (δ: 128.06), THF-$d_8$ (δ: 67.21); chemical shift values for phosphorus are relative to 85% H$_3$PO$_4$. NMR multiplicities are abbreviated as follows: s = singlet, d = doublet, t = triplet, q = quartet, sept = septet, m = multiplet, br = broad signal. Chemical shifts are quoted in δ (ppm) and coupling constants in Hz. The samples were dissolved in deuterated solvents, and were sealed off in J-Young NMR tubes for measurements. For the single crystal X-ray structure analysis, the crystals were each mounted on a glass capillary in perfluorinated oil and measured in a cold N$_2$ flow. The data for all compounds were collected on a Bruker D8 Venture or XtaLAB Synergy R, DW system, HyPix diffractometer at low temperatures.

## Data availability

All data generated or analyzed during this study are included in the Supplementary Information. Details about materials and methods, experimental procedures, characterization data, and theoretical calculations are available in the Supplementary Information. The structures of **1–8** in the solid state were determined by single-crystal Xray diffraction studies and the crystallographic data have been deposited with the Cambridge Crystallographic Data Center under nos. CCDC 2323602 (**1**), 2323603 (**2**), 2323604 (**3**), 2323605 (**4**), 2349216 (**6**), 2323606 (**7**), 2323607 (**8**), 2323608 (**9**), and 2323609 (**10**). These data can be obtained free of charge from The Cambridge Crystallographic Data Center via www.ccdc.cam.ac.uk/data request/ cif. All data are also available from corresponding authors upon request. Source data are present in this paper. Source data are provided with this paper.

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

## Acknowledgements

We thank the National Natural Science Foundation of China (Grants: 22322112 and 22071164, G.T.), Fundamental Research Funds for the Central Universities, Sun Yat-sen University (23qnpy35, G.T.), and the Suzhou Science & Technology NOVA Program (Grant: ZXL2022445, G.T.) for generous financial support. The computational work is supported by National Supercomputer Center (Tianhe-2 Supercomputer) in Guangzhou.

## Author contributions

G.T. designed the project. Y.C. carried out the experimental and part theoretical work. Z.K. instructed the mechanistic studies, and P.S. performed the studies. D.W. performed single-crystal X-ray diffraction studies. Y.C. and P.S. contributed equally to the work.

## Competing interests

The authors declare no competing interests.
