## [Peer Review File · Nature Communications]

Molecular-strain induced phosphinidene reactivity of a phosphanorcaradieneReviewers' Comments:

Reviewer #1:

Remarks to the Author:

This manuscript describes the synthesis of a sterically encumbered hydrindacene ligand and its use in stabilizing various interesting one-coordinate main group species and main group multiple bonds. The authors previously prepared triplet stibinidene and bismuthinidenes, intriguing low-valent Group 15 compounds. In this study, they synthesized a phosphanorcaradiene on the way to accessing free phosphinidene. The authors propose that phosphinidene is transiently generated during the reduction of the phosphorus dichloride precursor, forming a PC₂ three-membered ring through [2+1] cycloaddition with the flanking benzene ring's C=C double bond, illustrating the challenge of isolating free phosphinidene. They demonstrate that phosphanorcaradiene can serve as a synthetic equivalent of transient phosphinidene as a consequence of the high molecular strain of the three-membered ring. Overall, this manuscript not only sheds light on the unique properties and reactivity of phosphanorcaradiene but also opens up new avenues for further research into its potential applications in the synthesis of other phosphorus compounds. The findings presented here hold great promise and contribute to the field of phosphorus chemistry. Consequently, this reviewer suggests its publication in Nature Communications after addressing the following minor issues:

- 1) The authors should compare the reduction outcomes of phosphorus dichloride using the hydrindacene ligand and other ligand systems, as diphosphenes or higher oligomers are typically formed in other systems. An explanation for this difference should be provided.
- 2) Crystallographic analysis reveals two independent enantiomers in one crystal unit cell of compound 1; did the authors observe them in NMR spectroscopy?
- 3) The activation of Si-H and N-H bonds is notable. This reviewer suggests investigating reactions with dioxygen and dihydrogen or providing comments on potential outcomes.

Reviewer #2:

Remarks to the Author:

In this manuscript, Tan and coworkers are investigating the reactivity of phosphanorcaradiene with a series of nitrogen, phosphorus and carbon centred Lewis bases and with small molecules. They observed the Si-H, N-H bonds activation in silanes and amines and the 2 + 1 addition with alkenes and alkynes.

The interesting structures of the products have been investigated in details in the solid-state and the electronic structures were elucidated by quantum-chemical calculations.

This study will thus be of interest for a broad readership including chemistry working in inorganic, organometallic and main-group chemistry. This is a study of high scientific quality with a well presented manuscript and detailed analysis of the intriguing reactivity of ring-strained phosphorus derivatives.

In terms of novelty, the recent articles of Dobrovetsky and Radosevich (JACS, 2023, 145, 3786 and Chem Sci 2018 9 4338) have to be cited and similar reactivity pattern of geometrically constrained phosphines have already been observed.

I am really surprised that the steric, electronic and basic properties of the phosphines 1, 5 and 6 were not investigated, since this will make the paper interesting for a broader audience and reach the interest of chemists working in organometallic catalysis and coordination chemistry. Organic azide should also be probed for checking if these phosphines can be further used in organic chemistry such as in Staudinger and aza-Wittig reactions.

I would thus suggest to perform major revisions, put better the introduction in the context of geometrically constrained main-group compounds (Chem. Soc. Rev., 2024,53, 764-792,) and also

insert the additional reactivity studies described above.

Thus

Reviewer #3:

Remarks to the Author:

In their manuscript, Tan and co-workers report the synthesis and complete characterization of a novel phosphanorcaradiene unexpectedly formed in the reduction of a phosphorus(III) dichloride precursor with two molar equivalents of potassium graphite. This species constitutes, to the best of my knowledge, the second stable phosphanorcaradiene compound after the initial report by Stephan and co-workers in 2018 (JACS 2018, 140, 7466). In addition, the reactivity of this species towards Lewis bases, alkene/alkyne, and Si-H/N-H activations has been explored confirming the usefulness of this species as a synthetic equivalent of phosphinidene in small molecule activation. Moreover, DFT calculations were carried out to understand the bonding situation of this novel compound. Although this work is devoted only to one compound, it represents a significant step forward in the preparation of rare and challenging phosphorus compounds. As the manuscript is well-written and organized, and the work has been competently carried out, I support its publication in NatCommun.

Despite that, the following issues related to the computational part (my main area of expertise) should be addressed in a revision:

(i) According to the X-ray derived distances, the P-C bonds in the PC₂ ring are significantly longer than those reported by the analogous species reported by Stephan and coworkers. In addition, the C-C bond is comparatively shorter. This indicates that the C₂...P interaction in the title compound should be weaker than that in the analogous Stephan's phosphanorcaradiene. In my opinion, it would be helpful if the authors could discuss the differences between the title compound and the first reported phosphanorcaradiene (in terms of WBIs and AIM-delocalization indices).

(ii) In addition, the proposed [2+1] cycloaddition between the transient phosphinidene and the aromatic C=C bond should be computed to show the (kinetic) easiness of the process (despite the aromaticity loss).

(iii) For completeness, the profiles involving the activation of ethylene (or acetylene) and the Si-H (or N-H) activation reactions should be also computed to confirm the feasibility of these reactions at room temperature.

(iv) I do not really understand why the NBO calculations were carried out a completely different level than the geometry optimizations. A justification should be given.

(v) There is a typo in Figure S4 (Energy should be "Energy")

Reviewer 1:

Comment: This manuscript describes the synthesis of a sterically encumbered hydrindacene ligand and its use in stabilizing various interesting one-coordinate main group species and main group multiple bonds. The authors previously prepared triplet stibinidene and bismuthinidenes, intriguing low-valent Group 15 compounds. In this study, they synthesized a phosphanorcaradiene on the way to accessing free phosphinidene. The authors propose that phosphinidene is transiently generated during the reduction of the phosphorus dichloride precursor, forming a PC₂ three-membered ring through [2+1] cycloaddition with the flanking benzene ring's C=C double bond, illustrating the challenge of isolating free phosphinidene. They demonstrate that phosphanorcaradiene can serve as a synthetic equivalent of transient phosphinidene as a consequence of the high molecular strain of the three-membered ring. Overall, this manuscript not only sheds light on the unique properties and reactivity of phosphanorcaradiene but also opens up new avenues for further research into its potential applications in the synthesis of other phosphorus compounds. The findings presented here hold great promise and contribute to the field of phosphorus chemistry. Consequently, this reviewer suggests its publication in Nature Communications after addressing the following minor issues:

Response: Thank you very much for the nice comments and valuable suggestions.

Comment: The authors should compare the reduction outcomes of phosphorus dichloride using the hydrindacene ligand and other ligand systems, as diphosphenes or higher oligomers are typically formed in other systems. An explanation for this difference should be provided.

Response: We have added the description with corresponding references as follows: Contrastingly, similar reduction reactions with less sterically hindered ligands afforded diphosphenes or other higher oligomers.

Comment: 2. Crystallographic analysis reveals two independent enantiomers in one crystal unit cell of compound 1; did the authors observe them in NMR spectroscopy?

Response: We did not observe two sets of signals, most probably suggesting the enantiomers have a fast equilibrium in solution.

Comment: 3. The activation of Si-H and N-H bonds is notable. This reviewer suggests investigating

reactions with dioxygen and dihydrogen or providing comments on potential outcomes.

Response: We have tried the activations of dioxygen and dihydrogen, however inconclusive mixtures were afforded.

Reviewer 2:

Comment: In this manuscript, Tan and coworkers are investigating the reactivity of phosphanorcaradiene with a series of nitrogen, phosphorus and carbon centred Lewis bases and with small molecules. They observed the Si-H, N-H bonds activation in silanes and amines and the 2 + 1 addition with alkenes and alkynes. The interesting structures of the products have been investigated in details in the solid-state and the electronic structures were elucidated by quantum-chemical calculations. This study will thus be of interest for a broad readership including chemistry working in inorganic, organometallic and main-group chemistry. This is a study of high scientific quality with a well presented manuscript and detailed analysis of the intriguing reactivity of ring-strained phosphorus derivatives.

Response: Thank you very much for the nice comments and valuable suggestions.

Comment: In terms of novelty, the recent articles of Dobrovetsky and Radosevich (JACS, 2023, 145, 3786 and Chem Sci 2018 9 4338) have to be cited and similar reactivity pattern of geometrically constrained phosphines have already been observed.

Response: The references have been cited.

Comment: I am really surprised that the steric, electronic and basic properties of the phosphines 1, 5 and 6 were not investigated, since this will make the paper interesting for a broader audience and reach the interest of chemists working in organometallic catalysis and coordination chemistry.

Response: We thank the reviewer for the nice suggestions. We are currently working on using these phosphines as ligands in coordination chemistry. Since the steric and electronic studies are more suitable for coordination chemistry, we will include these studies in the following work.

Comment: Organic azide should also be probed for checking if these phosphines can be further used in organic chemistry such as in Staudinger and aza-Wittig reactions.

Response: The reactions with organic azide has been added in the text. **1** could react with one and two molar equivalents of 4-tertbutylphenylazide to afford iminophosphane and bis(imino)phosphane, respectively. However, **5** and **6** did not react with this azide.

Comment: I would thus suggest to perform major revisions, put better the introduction in the context of geometrically constrained main-group compounds (Chem. Soc. Rev., 2024,53, 764-792,) and also insert the additional reactivity studies described above.

Response: We are grateful for the reviewer for spending valuable time to evaluation our manuscript, and moreover giving us the nice comments and valuable suggestions to improve the quality of this manuscript. The mentioned references have been added.

Reviewer 3:

Comment: According to the X-ray derived distances, the P-C bonds in the PC₂ ring are significantly longer than those reported by the analogous species reported by Stephan and coworkers. In addition, the C-C bond is comparatively shorter. This indicates that the C₂ ••• P interaction in the title compound should be weaker than that in the analogous Stephan's phosphanorcaradiene. In my opinion, it would be helpful if the authors could discuss the differences between the title compound and the first reported phosphanorcaradiene (in terms of WBIs and AIM-delocalization indices).

Response: Thank you very much for the nice comments and valuable suggestions. The differences between **1** and Stephan's phosphanorcaradiene have been discussed in the text.

Comment: In addition, the proposed [2+1] cycloaddition between the transient phosphinidene and the aromatic C=C bond should be computed to show the (kinetic) easiness of the process (despite the aromaticity loss).

Response: Although the flanking fluorenyl group undergoes de-aromatization to form PC₂ ring, the calculated single-point energies show that the title compound has lower energy than the triplet phosphinidene which still keeps C=C double bond in the flanking fluorenyl group.

Comment: For completeness, the profiles involving the activation of ethylene (or acetylene) and the Si-H (or N-H) activation reactions should be also computed to confirm the feasibility of these reactions

at room temperature.

Response: The mechanisms have been commutated, please see the text for details.

Comment: I do not really understand why the NBO calculations were carried out a completely different level than the geometry optimizations. A justification should be given.

Response: The calculation level of NBO has been revised to BP86(D3BJ)/Def2SVP, which is at the same level as geometry optimizations.

Comment: There is a typo in Figure S4 (Enegrly should be “Energy”)

Response: This mistake has been revised.

Reviewers' Comments:

Reviewer #1:

Remarks to the Author:

The revised manuscript now not only meets the rigorous standards of the journal but also effectively addresses the concerns raised by the reviewers. Consequently, this reviewer believes the manuscript has reached a level of quality that warrants its acceptance for publication.

Reviewer #2:

Remarks to the Author:

The manuscript has been now modified by the authors, and most answers and revisions have been performed.

I would another time emphasize the importance of analyzing properly the steric and electronic properties of new phosphine such as the phosphanorcaradiene 1 reported in this work. Thus, the first part about the preparation and characterization of compound 1 and the theoretical calculations are missing information concerning the reactivity of the phosphorus atom and a simple calculation of a proton affinity and of one steric parameter (Tolman cone angle or buried volume) would be sufficient to compare rigorously the properties of this special phosphine with well-known phosphines known in the literature such as PPh₃ or analogous three-membered ring phosphirane Mes-P(CH₂CH₂).

Reviewer #3:

Remarks to the Author:

The authors have addressed most of the issues commented in my previous report in a satisfactory way. I therefore support the acceptance of the revised manuscript. However, the following (minor) issues should be considered:

- (i) It would be helpful for the readers if the computational level could be shown (at least, in the caption for Figures 7 and 8).
- (ii) The interaction energies, and not only the distortion energies, should be given in Figure 8.
- (iii) The original works by Bickelhaupt and Houk (ChemSocRev 2014,43, 4953 and AngewChemIntEd 2017, 56, 10070) on the distortion/interaction analysis should be cited in the main manuscript.
- (iv) Some references in the supporting information are confusing. For instance, S21 refers to the Mitoraj's NOCV method and has nothing to with the IBOs (J. Chem. Theory Comput. 2013, 9, 11, 4834). Please check it carefully and revise accordingly.

Reviewer 2:

Comment: The manuscript has been now modified by the authors, and most answers and revisions have been performed.

I would another time emphasis the importance of analyzing properly the steric and electronic properties of new phosphine such as the phosphanorcaradiene **1** reported in this work. Thus, the first part about the preparation and characterization of compound **1** and the theoretical calculations are missing information concerning the reactivity of the phosphorus atom and a simple calculation of a proton affinity and of one steric parameter (Tolman cone angle or buried volume) would be sufficient to compare rigorously the properties of this special phosphine with well-known phosphines known in the literature such as PPh₃ or analogous three-membered ring phosphirane Mes-P(CH₂CH₂).

Response: Thank you very much for the nice comments and valuable suggestions.

Calculations of proton affinity and buried volume for the phosphanorcaradiene **1**, PPh₃ and analogous three-membered ring phosphirane Mes-P(CH₂CH₂) have been performed. According to these calculation results, the proton affinity of **1** is better than those of PPh₃ and Mes-P(CH₂CH₂), indicating a better electron donating ability. The percent buried volume of **1** is larger than those of PPh₃ and Mes-P(CH₂CH₂), suggesting that the steric hindrance of **1** is greater than those of PPh₃ and Mes-P(CH₂CH₂). The steric effect guarantees the stability of this active species. Relevant results and discussion are given in the revised manuscript and shown below:

Figure 1. Analysis on proton affinity and steric effect of **1**, PPh₃ and Mes-P(CH₂CH₂).

“To provide further understanding of **1**, electronic and steric properties were theoretically studied. The proton affinity was analyzed to evaluate the electronic property, and meanwhile related calculations on PPh₃ and Mes-P(CH₂CH₂) were also performed for comparison. As shown in Supplementary Figure 5,

the proton affinity of **1** is -156.8 kcal/mol, which is lower than those of PPh_3 (-152.8 kcal/mol) and $\text{Mes-P}(\text{CH}_2\text{CH}_2)$ (-141.8 kcal/mol). Compared with PPh_3 and the analogous three-membered ring phosphirane $\text{Mes-P}(\text{CH}_2\text{CH}_2)$, **1** has a better capability of electron donating. In addition, the steric property of **1** were studied with percent buried volume ($\%V_{\text{bur}}$) analysis. The $\%V_{\text{bur}}$ of **1** is 76.6%, higher than those of PPh_3 (65.0%) and $\text{Mes-P}(\text{CH}_2\text{CH}_2)$ (54.3%). The results of percent buried volumes suggest that the steric hindrance around the phosphorus atom of **1** is greater than those of PPh_3 and $\text{Mes-P}(\text{CH}_2\text{CH}_2)$. Besides, steric maps shown in Supplementary Figure 5 also display a larger steric hindrance of **1** than PPh_3 and $\text{Mes-P}(\text{CH}_2\text{CH}_2)$. Owing to the steric effect, this active phophanorcaradiene species **1** can be stabilized.”

Reviewer 3:

Comment: The authors have addressed most of the issues commented in my previous report in a satisfactory way. I therefore support the acceptance of the revised manuscript.

Response: Thank you very much for the nice comments and valuable suggestions.

However, the following (minor) issues should be considered:

(i) It would be helpful for the readers if the computational level could be shown (at least, in the caption for Figures 7 and 8).

Response: This has been added.

(ii) The interaction energies, and not only the distortion energies, should be given in Figure 8.

Response: This has been added.

(iii) The original works by Bickelhaupt and Houk (ChemSocRev 2014,43, 4953 and AngewChemIntEd 2017, 56, 10070) on the distortion/interaction analysis should be cited in the main manuscript.

Response: These two references have been added.

(iv) Some references in the supporting information are confusing. For instance, S21 refers to the Mitoraj's NOCV method and has nothing to with the IBOs (J. Chem. Theory Comput. 2013, 9, 11, 4834). Please check it carefully and revise accordingly.

Response: The references in the supplementary information have been revised.